# The Effects of Curcumin on Inflammasome: Latest Update

**DOI:** 10.3390/molecules28020742

**Published:** 2023-01-11

**Authors:** Tarek Benameur, Socorro Vanesca Frota Gaban, Giulia Giacomucci, Francesca Martina Filannino, Teresa Trotta, Rita Polito, Giovanni Messina, Chiara Porro, Maria Antonietta Panaro

**Affiliations:** 1College of Medicine, Department of Biomedical Sciences, King Faisal University, Al-Ahsa 31982, Saudi Arabia; 2Department of Food Engineering, Federal University of Ceara, Campus do Pici., Fortaleza CEP60356-000, Brazil; 3Department of Neuroscience, Psychology, Drug Research and Child Health, University of Florence, 50134 Florence, Italy; 4Department of Clinical and Experimental Medicine, University of Foggia, 71121 Foggia, Italy; 5Department of Biosciences, Biotechnologies and Environment, University of Bari, 70125 Bari, Italy

**Keywords:** curcumin, natural flavonoid, inflammasome, neuroinflammation, anti-inflammatory, diseases, lung diseases, arthritic diseases

## Abstract

Curcumin, a traditional Chinese medicine extracted from natural plant rhizomes, has become a candidate drug for the treatment of different diseases due to its anti-inflammatory, anticancer, antioxidant, and antibacterial activities. Curcumin is generally beneficial to improve human health with anti-inflammatory and antioxidative properties as well as antitumor and immunoregulatory properties. Inflammasomes are NLR family, pyrin domain-containing 3 (NLRP3) proteins that are activated in response to a variety of stress signals and that promote the proteolytic conversion of pro-interleukin-1β and pro-interleukin-18 into active forms, which are central mediators of the inflammatory response; inflammasomes can also induce pyroptosis, a type of cell death. The NLRP3 protein is involved in a variety of inflammatory pathologies, including neurological and autoimmune disorders, lung diseases, atherosclerosis, myocardial infarction, and many others. Different functional foods may have preventive and therapeutic effects in a wide range of pathologies in which inflammasome proteins are activated. In this review, we have focused on curcumin and evidenced its therapeutic potential in inflammatory diseases such as neurodegenerative diseases, respiratory diseases, and arthritis by acting on the inflammasome.

## 1. Introduction

Curcumin, a natural monomer extracted from plants, has gained popularity in recent decades due to its therapeutic benefits in a wide range of human pathological conditions. The medicinal plant *Curcuma longa Linn*, a perennial herb of the *Zingiberaceae* family known as “golden spice” for its broad spectrum of pharmacological properties, contains curcumin as one of its most active constituents [1].

Curcumin, with a chemical formula of $C_{21}H_{20}O_{6}$ and molecular weight of 368.38 g/mole, is also known by its IUPAC name (1E,6E)-1,7-bis (4-hydroxy-3-methoxyphenyl)-1,6-heptadiene-3,5-dione 1. It is a polyphenolic compound derived from the rhizome of the herb *Curcuma longa*, and it has been shown to have antioxidative, anti-inflammatory, anticancer, chemopreventive, and anti-neurodegenerative properties [2]. Curcumin’s anti-inflammatory properties have been supported by numerous preclinical and clinical studies in a variety of inflammatory diseases.

Due to its anti-inflammatory and antioxidant properties, curcumin is considered a promising drug candidate for the treatment of inflammatory diseases. Inflammation is the body’s response to certain stimuli, such as viral or bacterial infections, mechanical damage, or an excessive immune response. The purpose of the inflammatory response is to remove stimulating factors and accelerate tissue repair, but excessive inflammatory responses can lead to tissue damage, organ dysfunction, and potentially life-threatening disease.

In addition, the chemical structure of curcumin makes it an excellent scavenger of reactive oxygen and nitrogen species (ROS and RNS, respectively) [3]. As a result, curcumin can attenuate or prevent exercise-induced oxidative stress and inflammation [4,5]. Curcumin activates the Nrf2 pathway, which is important for the activation of antioxidant enzymes such as thioredoxin reductase, Hsp70, and sirtuins [6].

Curcumin’s therapeutic activity is limited by its poor water solubility, fast biological metabolism, and low bioavailability due to insufficient absorption, chemical instability, and rapid metabolism in the body, which is one of the limitations of its use as a potential therapeutic agent [7]. One approach to addressing these issues is through the use of nanocarriers, which are drug delivery systems optimized for improving curcumin bioavailability and thus its therapeutic effects. It activates caspase-1, which results in the processing and secretion of critical inflammatory molecules.

Due to their nanometric size and chemical properties, nanoparticles, liposomes, micelles, phospholipid vesicles, and polymeric nanoparticles can increase the effectiveness of curcumin. Exosomes are extracellular vesicles that are normally released from cells by exocytosis after multivesicular bodies have matured [8]. They are natural nanocarriers for drug delivery [9]. 

The NLR family, pyrin domain-containing 3 (NLRP3) protein is an intracellular signaling molecule that recognizes a wide range of pathogens, environmental factors, and host-derived factors, including ATP [10]. The NLRP3 inflammasome consists of the NLRP3, the ASC adaptor, and caspase-1, which is activated by various danger signals [11]. When activated, it activates caspase-1, which leads to the processing and secretion of critical inflammatory molecules such as the pro-inflammatory cytokines IL-1β and IL-18 [12].

Normal activation of the NLRP3 inflammasome contributes to host defense, but abnormal activation is pathogenic in inherited disorders such as cryopyrin-associated periodic syndrome (CAPS) and complex diseases such as multiple sclerosis, type 2 diabetes, Alzheimer’s disease, and atherosclerosis [13]. 

Both inflammasome formation and its activity play a critical role in a variety of pathologies, including cardiovascular, metabolic, renal, digestive, and central nervous system (CNS) diseases.

The aim of this review is to elucidate the role of curcumin in inducing protection by regulating inflammasomes in a variety of diseases.

## 2. Inflammasome

Inflammasomes are multimeric protein complexes that are widely expressed in the cytoplasm of various cell types, including immune and non-immune cells [14]. Inflammasomes mediate the host’s innate immune responses to microbial infection and cellular damage. Cells are activated through the intervention of germline-encoded pattern-recognition receptors (PRRs) on the surface of the cell membrane. Pathogen-associated molecular patterns (PAMPs) are recognized by PRRs. Examples of conserved microbial factors detected by PRRs include the bacterial secretion system, microbial nucleic acids, and elements of the microbial cell wall. PRRs are also triggered by endogenous danger-associated molecular patterns (DAMPs) generated in the setting of cellular injury or tissue damage [15], such as ATP, uric acid crystals, heat-shock proteins hsp70 and hsp90, and the high-mobility group box 1 (HMGB1) [16]. The binding between PRRs and PAMPs or DAMPs induces NF-κB activation and the expression of the inflammasome [17]. Inflammasome protein complexes are classified into canonical and non-canonical. The classification depends on the activation of cysteinyl and the mode of activation of cysteinyl aspartate specific proteinase (Caspase) during inflammasome formation. Canonical inflammasomes are constructed by the nucleotide-binding oligomerization domain (NOD) leucine-rich repeat (LRR)-containing protein receptors (NLRs). The canonical inflammasomes include Nlrp3, Nlrp1b, Nlrc4, the ALR member absent in melanoma 2 (AIM2), and pyrin. [18]. These assemble canonical inflammasomes that promote activation of the cysteine protease caspase-1 (Figure 1). The non-canonical inflammasome promotes activation of procaspase-11 (caspase-4 and caspase-5 in human). Recently, extracellular lipopolysaccharide (LPS) has been found to trigger the activation of the non-canonical inflammasome. LPS induces the expression of pro-IL-1β and NLRP3 via the TLR4-MyD88-dependent pathway and type I interferon via the TLR4-TRIF-dependent pathway. Type I interferon results in a feedback loop and activates type I interferon receptor (IFNAR) to induce caspase-11 expression [19].

Additionally, AIM2 is a protein that is the founding member of the AIM2-like receptor (ALR) family. ALRs are typically bipartite proteins with one or more HIN domains following the N-terminal pyrin domain (PYD). Caspase-1 is mainly activated by inflammasome assembly. Nucleotide-binding domain and leucine-rich repeat receptors (NLRs) or absent in melanoma 2 (AIM2)-like receptors (ALRs) initiate the assembly of the canonical inflammasome complex [14]. NLRs and ALRs mediate host recognition of PAMPs released during infection or DAMPs released during cell damage. In the case of some of these PRRs, such as NLRP1, NLRP3, and AIM2, the skeleton protein consists mainly of a carboxyl-terminal containing leucine-rich repeats (LRRs) that recognizes the ligand PAMPs. A central domain includes the NACHT cassette, which hydrolyzes adenosine triphosphates (ATP) by activating a deoxyribonucleoside triphosphate (dNTP) enzyme, and an amino acid terminal domain consisting of caspase-recruitment domains (CARD), a pyrin domain (PYD), an acid transactivator, and baculovirus inhibitor repeats (BIRs).

Two signals are usually required for the activation of the canonical inflammasome. Ligands such as PAMPs and DAMPs that activate inflammasomes make up the second signal. PAMPs and DAMPs are recognized by the host through the action of NLRs and ALRs. The N-terminal CARD facilitates the recruitment of pro-caspase-1 to the inflammasome complex by attracting apoptosis-associated speck-like protein, which contains a caspase-recruitment domain (ASC) [17]. A subset of these receptors subsequently assembles cytosolic protein complexes known as inflammasomes, which activate proinflammatory caspase-1 and -11. In most cases, activated NLRs and ALRs recruit a bipartite protein ASC to engage caspase-1 activation. In macrophages or dendritic cells, inflammasome-forming NLRs and ALRs induce the reorganization of cytoplasmic ASC into a single ‘speck’ of 0.8–1 μm, which is thought to be a hallmark of inflammasome assembly. ASC is critical for the recruitment of caspase-1 to facilitate the proteolytic processing of pro-IL-1β and pro-IL-18.

In fact, upon activation of inflammasomes, only caspase-1 produces mature IL-1β, a pyrogenic cytokine that also promotes adaptive T helper 1 (Th1), Th17, and humoral immunity. IL-1β is an important proinflammatory mediator that is generated at sites of injury or immunological challenges to coordinate programs as diverse as a cellular recruitment to a site of infection or injury. In addition, caspase-1 produces mature IL-18, which is important for IL-17 expression by Th17 cells and, when combined with other cytokines, can polarize T cells towards Th1 or Th2 profiles. Finally, caspase-1-processed IL-33 mainly induces Th2 cells to release IL-13 and IL-5 [20]. In addition, the activation of caspase-1 and caspase-11 triggers the cleavage of the gasdermin D protein (GSDMD). GSDMD splits into two fragments: the N-terminal domain and the C-domain. The N-terminal domain of GSDMD (N-GSDMD) oligomerizes and forms plasma membrane pores on lipid membranes and induces pyroptosis that mediates cell death and the secretion of mature forms of IL-1β and IL-18 [14]. Pyroptosis is a non-homeostatic and lytic mode of cell death that requires caspase-1 or -11 enzymatic activities. Like caspase-independent necroptotic cell death, pyroptotic cell death is characterized by cytoplasmic swelling and plasma membrane rupture [16,21]. Excessive pyroptosis may promote sepsis by inducing immunosuppression while amplifying the inflammatory response.

NLRP1 was the first member of the NLR family identified to form an inflammasome complex [11]. Humans have a single NLRP1 gene, while the mouse genome encodes three paralogs: Nlrp1a, Nlrp1b, and Nlrp1c. The Nlrp1 inflammasome is an important defense mechanism against *Bacillus anthraces*, as this variant responds to lethal toxins. Nlrp1b is the key locus that determines whether macrophages undergo pyroptosis and IL-1β secretion in response to bacterial toxins. Toxins and muramyl dipeptide (MDP) as PAMPs lead to an efflux of intracellular potassium ions (K^+^), activate the NLRP1 inflammasome, and induce IL-1β secretion by macrophages.

The NLRP3 inflammasome is currently the best-characterized inflammasome. NLRP3 is activated when exposed to a wide range of PAMPs and DAMPs, such as bacterial messenger RNAs, bacterial DNAs, MDP, DNA and RNA viruses, and several host-derived molecules indicative of injury, including extracellular ATP released by injured cells.

The activation of the NLRP3 inflammasome requires two separate signals. Signal one (priming process) is triggered by activation of Toll-like receptor (TLR) 4 or tumor necrosis factor (TNF) signaling, which subsequently leads to the transcriptional activation of NLRP3. TLRs, IL-1Rs, and TNFRs induce transcription and translation of NF-κB to produce proforms of IL-1β and IL-18 [22]. As a second step, enzymatic cleavage is required to secrete these pro-inflammatory ILs into the extracellular space. This requires oligomerization of NLRP3 with ASC, facilitating proteolytic cleavage of pro-caspase-1 to its active caspase-1. Caspase-1 then cleaves the pro-cytokines into mature IL-1β and IL-18, which can be secreted. However, recent studies have shown that caspase-11 (caspase-4 and caspase-5 in humans) is required for the non-canonical activation of caspase-1 in macrophages infected with the enteric bacteria Escherichia coli, Citrobacter rodentium, and Vibrio cholerae, whereas caspase-11 is dispensable for caspase-1 activation by canonical NLRP3 activators such as ATP and nigericin.

Furthermore, the activation of the NLRP3 inflammasome in LPS-induced endotoxaemia is caspase-11 dependent. The activation signal two (protein complex assembly) is induced by various pathogen-associated molecular patterns (PAMPs) and damage-associated molecular patterns (DAMPs), including extracellular ATP, pore-forming toxins, RNA viruses, and particulate matter. Interleukin 1β (IL-1β)/IL-1R1, LPS/Toll-like receptor 4 (TLR4), tumor necrosis factor (TNF)/TNF receptor (TNFR), sphingosine 1-phosphate (S1P)/S1P receptor 2 (S1PR2), adenosine diphosphate (ADP)/P2Y12, α-synuclein/CD36, and bromodomain-containing protein 4 (BRD4) inhibitor. JQ1 activates NF-κB and then upregulates the transcription of the component required for NLRP3 inflammasome formation. NLRP3 interacts with a wide range of activators. It is unlikely that NLRP3 interacts physically with its activators. NLRP3 may instead detect a common cellular signal induced in response to NLRP3 activators [23]. 

The identity of this signal is currently under intensive debate. Several molecular and cellular events, including K^+^ efflux, Ca^2+^ signaling, reactive oxygen species (ROS), mitochondrial dysfunction, and lysosomal rupture have been proposed as the trigger(s) for NLRP3 inflammasome activation, [22]. Several studies showed that nigericin, a potassium ionophore, stimulates IL-1β maturation in LPS-stimulated murine macrophages. Furthermore, nigericin promotes K^+^ efflux, and a decrease in cytosolic K^+^ concentration is sufficient to activate the NLRP3 inflammasome. Furthermore, studies showing that nigericin induces Ca^2+^ mobilization during the NLRP3 inflammasome activation process [23] support the involvement of Ca^2+^ signaling in NLRP3 inflammasome activation. NLRP3 has been shown to reside in the endoplasmic reticulum (ER) and cytosol. 

Following activation by various stimuli, the NLRP3 inflammatory complex is assembled at MAMs, which are highly specialized contact sites between the ER and mitochondria. The localization of NLRP3 to MAMs/mitochondria may contribute to the immediate recognition of mitochondrial damage, mitochondrial DNA (mtDNA) translocation, and cardiolipin, followed by a response. It was recently proposed that mtDNA released into the cytosol is the “ultimate” trigger of the NLRP3 inflammasome under various “priming” and “activation” conditions [24]. The co-localization of NLRP3 and cytoplasmic mtDNA induces the production of IL-1β. In the cytosol, the presence of oxidized mtDNA is a more potent inducer of IL-1β secretion [24]. Importantly, NLRP3 localization to the MTOC leads to its interaction with the centrosome-localized mitotic kinase NEK7, allowing NLRP3 inflammasome assembly [24]. A crucial protein is the mitotic serine/threonine-protein kinase NEK7.

NEK7 interacts with NLRP3, which causes oligomerization of NLRP3 [25]. A recent structural modeling study revealed that the molecular mechanism of NEK7–NLRP3 interactions bridges adjacent NLRP3 subunits and promotes NLRP3 inflammasome oligomerization. It is unclear whether NEK7 is absolutely required for NLRP3 oligomerization and further inflammasome assembly [25]. Recent research has highlighted the importance of the Golgi apparatus and its lipid mediators in the aggregation of NLRP3 and the activation of NLRP3 inflammasome assembly [25]. Indeed, when exposed to specific stimuli, NLRP3 causes the trans-Golgi network (TGN) to disassemble into the dispersed TGN (dTGN). Notably, K^+^ efflux-independent stimuli cause NLRP3-dTGN activation, which leads to aggregation and activation of the NLRP3 inflammasome [25]. In addition, NLRP3 inflammasome activation is dependent on the ER-to-Golgi translocation of sterol regulatory element-binding protein (SREBP) 2 and SREBP cleavage-activating protein (SCAP), which form a ternary complex with NLRP3. Another recent study found that the NLRP3 inflammasome causes MAMs to be localized near Golgi membranes. This inter-organelle communication is dependent on the recruitment of protein kinase D (PKD) to DAG sites at the Golgi, which facilitates NLRP3 oligomerization and assembly of the active inflammasome [25]. 

Intense efforts have been made over the past decade to investigate the mechanism of NLRP3 inflammasome activation.

The identification of an alternative NLRP3 inflammasome signaling pathway is important. Recent studies have shown that LPS induces an “alternative inflammasome” in human monocytes. This alternative inflammasome activation was propagated by TLR4-TRIF-RIPK1-FADD-CASP8 signaling upstream of NLRP3 [23]. In this model, LPS induces the release of endogenous ATP from human monocytes, which in turn activates the P2X7 receptor to trigger NLRP3 inflammasome activation and IL-1β maturation. This pathway does not require K^+^ efflux, which is in contrast to NLRP3 activation induced by ATP, pore-forming toxins and particulate matter. Therefore, this pathway is defined as an alternative NLRP3 inflammasome pathway. After LPS treatment, the molecules RIPK1, FADD, and CASP8 act downstream of TLR4-TRIF signaling to activate NLRP3. 

Although both ASC and caspase-1 are required, there is no evidence of ASC speck formation or pyroptosis in this pathway [23]. The NLRC4 inflammasome is normally activated by two crucial components of pathogenic bacteria: flagellin of Gram-positive and Gram-negative bacteria, and proteins from the type III secretion system (T3SS) proteins, which originate from Gram-negative bacteria and inject virulence factors into the host cell [26]. The ALR protein recruits AIM2 to form a conventional inflammasome, which then activates caspase-1. Oligosaccharides and PYDs make up the structure of the AIM2 inflammasome. The AIM2 inflammasome consists of oligosaccharide domains and PYDs. The AIM2 inflammasome identifies double-stranded DNA in the cell using the oligosaccharide domain and then recruits ASC via the PYD, which leads to self-activation of pro-caspase-1. AIM2 has been implicated in the pathology of autoimmune disorders such as systemic lupus erythematosus, in which anti-DNA autoantibodies are abundant. If the pathway is confirmed, it will be a candidate for targeted therapies [17]. The pyrin protein is a 781-amino acid, ~95 kDa protein encoded on chromosome 16 by MEFV. Pyrin is expressed by innate immune system cells such as monocytes, granulocytes, and eosinophils. Homology analyses revealed five distinct domains within the pyrin protein. The eponymous PYD (1–92) is found at the N-terminus of pyrin and is found in more than 20 human proteins that are primarily involved in inflammatory processes. The PYD binds apoptosis-associated speck-like protein with a caspase-recruitment domain (ASC), resulting in caspase-1-mediated IL-1β production. 

The C-terminal B30.2 domain of pyrin is critical for the molecular mechanisms leading to familial Mediterranean fever (FMF). Most of the FMF-associated mutations are clustered in this C-terminal domain. Familial Mediterranean fever (FMF) is associated with mutations in the MEFV gene. In addition, pyrin-associated auto-inflammation with neutrophilic dermatosis (PAAND) has been associated with MEFV mutation. Recent findings have now revealed how pyrin is activated during infection. FMF is a prototypic auto-inflammatory disease characterized by recurrent episodes of fever with serosal inflammation manifesting with severe abdominal or chest pain, arthralgia, monoarticular arthritis, and limited erythematous skin rash. Pyrin does not directly identify molecular patterns (pathogen- or host-derived danger molecules) [27]. Rather, pyrin responds to perturbations in cytoplasmic homeostasis caused by the infection. These disturbances, recently defined as ‘homeostasis-altering molecular processes’ (HAMPs), are processes that result in RhoA GTPase inactivation [27,28].

## 3. Curcumin, Inflammasome, and Brain

Several brain disorders are characterized by an abnormal inflammatory response called neuroinflammation. In this process, glial cells, endothelial cells, and peripherally derived immune cells produce cytokines, chemokines, ROS, and second messengers that could disrupt the blood–brain barrier, resulting in cell damage and loss of neuronal function [29].

Curcumin is well known to have anti-inflammatory effects, and its role in modulating neuroinflammation in several neurological disorders has been extensively described [30].

Interestingly, emerging clinical and experimental evidence supports the hypothesis that neuroinflammation may play a role in the pathophysiology of epilepsy [31,32]. More specifically, it has been observed that IL-1β is consistently elevated in the hippocampus of experimentally induced epileptic animal models. The inflammasome regulates IL-1β maturation in a caspase-1-dependent manner. Caspase-1 is subsequently activated as a result of the NLRP3 inflammasome’s interaction with the inflammasome-adaptor protein ASC [33].

Taking these premises into account, it was hypothesized that the NLRP3 inflammasome might play an important role in the genesis of neuroinflammation, which contributes to the pathophysiology of epilepsy. Because curcumin has a well-defined anti-inflammatory role, He et al. investigated the effect of curcumin on the Kainic Acid (KA)-induced epilepsy model. Interestingly, the authors showed that IL-1β-activated caspase-1 and NLRP3 inflammasome were significantly upregulated in the hippocampus of epileptic rats along with significant neuronal damage.

Furthermore, when rats were treated with curcumin, there was a significant reduction in IL-1β-activated caspase-1 and NLRP3 expression as well as a reduction in neuronal loss when compared with untreated epileptic rats. Finally, curcumin-treated epileptic rats improved in tasks assessing spatial learning and memory. In view of these results, He et al. proposed that curcumin inhibited NLRP3 inflammasome activation and subsequent neuroinflammation in KA-induced epileptic animal models [34]. The link between epilepsy and inflammasome-activated neuroinflammation may be due to seizure activity that stimulates voltage-gated and NMDA-dependent channels, leading to an increase in channels and increased intracellular calcium and ROS production. Indeed, ROS can lead to the dissociation of thioredoxin-interacting protein (TXNIP) from oxidized thioredoxin-1 (Trx-1) and then to the activation of NLRP3 through the association of TXNIP with NLRP3 itself [35]. Given this evidence, curcumin has been suggested as a potential candidate for epilepsy therapy, although clinical trials are needed to confirm this hypothesis.

Ischemic brain damage can also result in neuroinflammation. Indeed, poststroke neurodegeneration frequently leads to brain inflammation, which sets off a cascade of events that results in secondary injury and white matter disruption [36]. White matter injury is characterized by demyelination and axonal integrity loss. Several studies have highlighted that it is a major cause of long-term deficits and cognitive decline [37].

The inflammatory response mediated by microglia plays a significant role in the secondary damage of white matter following stroke [38]. Indeed, following the acute phase of an ischemic stroke, microglia polarized towards a proinflammatory type that secreted proinflammatory cytokines and increased oligodendrocyte cells death and demyelination, leading to a worsening of white matter injury [39].

In addition, microglial pyroptosis, a type of pro-inflammatory programmed cell death, also plays an important role in neuroinflammation in stroke [40]. Pyroptosis is driven by inflammasome activation. In particular, it is initiated by NLRP3 activation and executed by GSDMD. More specifically, ischemic stroke results in the translocation of p50 and p65 into the nucleus of microglia, leading to the activation of NF-κB and consequently triggering the transcription of target genes such as gasdermin D (GSDMD), NLRP3, IL-1β, and IL-18. The NLRP3 inflammasome is then activated by oligomerization with ASC and pro-caspase-1, which activates caspase-1, which then cleaves the GSDMD, pro-IL-1β, and pro-IL-18 into GSDMD-N, IL-1β, and IL-18, respectively. GSDMD-N forms membrane pores after translocation to the plasma membrane, resulting in the release IL-1β and IL-18 and mediating pyroptosis [41]. Recent research suggests that NLRP3-mediated microglial pyroptosis may play an important role in post-stroke neuroinflammation [42].

Interestingly, curcumin has been shown to protect against stroke-induced neuronal damage by modulating microglial polarization [41,43]. Recently, in vivo and in vitro curcumin treatment has been shown to reduce stroke-induced white matter damage while attenuating microglial pyroptosis. Indeed, curcumin inhibits NLRP3 activation induced by ischemic stroke in animal models of transient middle cerebral artery occlusion/reperfusion (MCAO/R) via an upstream regulatory mechanism. Curcumin appears to suppress NF-κB signaling, which inhibits of NF-κB-mediated NLRP3 activation (Figure 1), thus directly acting as a regulator of microglial pyroptosis via this mechanism. This results in a reduction in white matter damage caused by ischemic stroke [41].

Furthermore, curcumin influences other aspects of the disease process. This includes microglial activity, antioxidant activity, and neuroinflammation via NLRP3 inhibition.

Curcumin has been described as a natural compound with potential efficacy in the “treatment” of AD [44], as it targets both β-amyloid (Aβ) and phosphorylated tau, the two hallmarks that drive neurodegeneration [45]. Curcumin also affects other aspects of the disease process. Notably, it modifies microglial activity, antioxidant activity, and modulates neuroinflammation via NLRP3 inhibition.

Indeed, Ruan, Y et al. developed a nano theranostic platform composed of curcumin and superparamagnetic iron oxide (SPIO) nanoparticles encapsulated by diblock 1,2-dio-leoyl-sn-glycero-surface (SDP@Cur-CRT/QSH). They demonstrated that this nanomaterial reduced Aβ plaques in APP/PS1 transgenic mice.

Furthermore, SDP@Cur-CRT/QSH attenuated microglial activation and reduced Aβ-associated inflammation by directly targeting NLRP3. Indeed, NLRP3 and Aβ have a complex interaction; NLRP3 colocalizes with Aβ and tends to concentrate around Aβ plaques. The deposition of Aβ promotes neuroinflammation by activating the NLRP3 inflammasome in microglia, which is essential for the production of pro-inflammatory factors such as IL-1β and IL-18 [46,47]. Interestingly, in transgenic mice, SDP@Cur-CRT/QSH nanoparticles reduced IL-18, CD68 (a marker for activated microglia), and NLRP3, and this effect appeared to be mediated through a direct action on the NLRP3 pathway. Indeed, SDP@Cur-CRT/QSH significantly reduced both NLRP3 and one of its products (apoptosis-associated speck-like protein containing a CARD) in transgenic mouse hippocampal and cortical neurons. Based on their findings [48], they hypothesized that SDP@Cur-CRT/QSH nanoparticles specifically inhibit NLRP3 in microglia and neuron-associated inflammasomes, thereby reducing Aβ-induced neuroinflammation [48].

In addition to NLRP3, other inflammasomes have been implicated in neurological diseases. Specifically, several studies have shown that the NALP1 spinal inflammasome appears to be directly involved in the pathogenesis of neuropathic pain [49], suggesting a significant contribution of NALP1 to pain sensitization [50]. Indeed, in animal models of nerve injury (SNI)-induced neuropathic pain, NALP1 was activated in spinal astrocytes, although the precise pathogenic mechanics have not yet been fully understood. When activated, NALP1 induced a caspase-1-dependent cleavage of pro-IL-1β to form IL-1β, which has been strongly linked to the pathogenesis of neuropathic pain [51].

Interestingly, Liu et al. demonstrated that repeated curcumin treatment reduced NALP1 levels in spinal astrocytes, suppressing the neuropathic-pain-induced activation of this inflammasome and, consequently, the production of IL-1β [52]. Taken together, curcumin may be effective in relieving neuropathic pain by downregulating IL-1β, thereby suppressing the activation of the NALP1 inflammasome.

## 4. Curcumin, Inflammasome, and Lung

The NOD, LRR, and pyrin domain-containing protein 3 (NLRP3) is an intracellular sensor that detects a variety of microbial motifs, endogenous harmful signals, and environmental irritants. These result in the formation and activation of the NLRP3 inflammasome. This includes bacterial lung infections and pulmonary diseases [53]. Despite the powerful inhibitory effects of various pharmacological agents on the release of pro-inflammatory factors involved in acute lung injury (ALI), the efficacy of the currently available anti-inflammatory drugs for ALI remains poor [54].

Curcumin has been shown to have broad anti-inflammatory properties in vitro and in vivo. Given the importance of inflammasomes in the pathogenesis of many lung diseases associated with destructive inflammation, here we emphasize the latest updates about curcumin’s modulatory effects on inflammasome-related components in the pathogenesis of acute and chronic lung diseases.

Investigation of the protective effects of curcumin on ALI models has shown that curcumin reduced histopathological injury, lung inflammation, and myeloperoxidase (MPO) activity, which is used to measure neutrophil presence and as an indirect lung injury indicator [55]. Furthermore, serum levels of some inflammatory markers such as CCL7, IL-6, and TNF-α were reduced.

On the one hand, curcumin increased cell viability and increased pyroptosis-related protein expression in RAW256.7 cells. On the other hand, inflammation in these cells was reduced. Interestingly, the mortality rate of mice treated with curcumin was reduced. Remarkably, curcumin inhibited the inflammatory process by silencing NLRP3 inflammasome-dependent pyroptosis, increasing SIRT1, and decreasing GSDMD-N expression [56].

Similar effects have been observed in other ALI models such as paraquat-induced ALI. Paraquat (PQ) (N, N′-dimethyl-4, 4′-bipyridinium dichloride) is a widely used herbicide known for its high toxicity to the human body and for its ability to induce ALI through increased oxidative stress and inflammation [57].

Activation of the NLRP3 inflammasome plays an important role in PQ-induced lung injury [58]. As a result, NLRP3 has emerged as a promising therapeutic target for the treatment of PQ-induced ALI. Although curcumin significantly reduced PQ-induced ROS production and apoptosis. It did not fully protect lung cells from PQ-induced damage. After treating lung cells with curcumin, TXNIP and NLRP3-mediated proinflammatory cytokine release, including factors such as IL-1β and IL-18, was significantly reduced compared to PQ treatment alone, while the effects on cleaved caspase-1 levels were only slightly attenuated. Curcumin significantly attenuated Notch1 without affecting ERK1/2 phosphorylation.

These findings demonstrated that curcumin’s inhibitory effects on TXNIP suppresses NLRP3 inflammasome activation and consequently improves PQ-induced ALI [59]. Recently, researchers have focused on the protective effects of a water-soluble curcumin formulation on acute respiratory distress syndrome (ARDS), the most severe form of ALI, which is associated with reduced lung compliance and hypoxemia.

Curcumin administration significantly reduced mortality, lung injury, pathogen presence in the lungs and blood, oxidative stress, and inflammation in a mouse model of lethal Gram-negative pneumonia.

Following curcumin administration, the klebsiella hemolysin gene, TNF-α, IFN-β, nucleotide-binding domain, leucine-rich-containing family, pyrin-3, hypoxia-inducible factor 1/2α, and NF-κB were all reduced. Lung compliance was also improved. This suggests that curcumin reduced the severity of pneumonia by modulating the inflammasome complex and hypoxia signaling pathways [60].

Inflammasome activation is strongly linked to a number of pathophysiological conditions, including airway inflammation. *Curcuma phaeocaulis* extract was shown to suppress caspase-1 activation and pro-IL-1β maturation in SiO_2_- and TiO_2_-nanoparticle-activated NLRP3 inflammasome in macrophages. IL-1β levels were reduced in bronchoalveolar lavage (BAL) fluids from animals previously treated with nanoparticles and treated in vitro with *C. phaeocaulis*. *C. phaeocaulis* has an anti-inflammatory effect against lung disease associated with excessive activation of the NLRP3 inflammasome/inflammation [61,62].

These findings support previous research findings that demonstrate curcumin’s anti-inflammatory effects in a variety of pathological contexts, including pulmonary diseases and ALI. In this context, curcumin may be a promising treatment candidate.

Furthermore, increased inflammation is clearly linked to asthma exacerbation and airway remodeling. The discovery of NLRP3 as a novel therapeutic target for the most effective management of severe asthma exacerbation has prompted researchers to investigate the natural compound curcumin, which is known for its anti-inflammatory properties. LPS exposure activates the NLRP3 inflammasome via the Toll-like receptor 4 (TLR4), as previously reported [63].

Asthma is one of the most common chronic respiratory disorders. Severe asthma is characterized by recurrent attacks of reversible airflow obstruction, airway hyperresponsiveness (AHR) to environmental allergens, and chronic airway inflammation that is resistant to current treatments [64]. Asthma is an inflammatory disease of the airways associated with altered lung structure and airway remodeling, which frequently increase the severity of the disease.

Recent evidence suggests that NLRP3 activation is not only a key component of the innate immune response in airways but that it is also involved in asthma pathogenesis and is associated with asthma exacerbation and severity. Indeed, NLRP3 inflammasome assembly leads to caspase-1-dependent release of the pro-inflammatory cytokines IL-18 and IL-1β as well as pyroptosis [65]. Despite its role in pathogen clearance in the lung, persistent NLRP3 activation by environmental allergens and/or inhaled irritants has been shown to increase pulmonary inflammation and to lead to asthma exacerbation. In animal models of asthma, NLRP3 inhibitors are still ineffective in controlling AHR and pulmonary inflammation. Jaiswal et al. demonstrated that intranasal administration of curcumin in combination with dexamethasone modulates NLRP3 activation and regulates LPS-induced asthma exacerbation. These combined strategies are effective in reducing dexamethasone-induced side effects. The most obvious finding to emerge from this study is a significant reduction in inflammatory cell recruitment and restoration of structural changes in the lungs as well as altered expression of TLR-4, NF-κB, NLRP3, Caspase-1, IL-1β, MMP-9, IL-5, and IL-17 in groups treated with intranasal curcumin alone or in combination with corticosteroid [66]. The most common symptoms of chronic asthma are mucus hypersecretion and airway inflammation. Curcumin was found to attenuate pulmonary alterations in an ovalbumin (OVA)-induced chronic asthma model. In addition, the increased levels of the inflammatory mediators TNF-α, IL-4, IL-5, and IL-13 in OVA-induced chronic asthma were found to be significantly reduced in BALF. Similarly, curcumin administration reduced the number of inflammatory cells such as eosinophils, neutrophils, macrophages, and lymphocytes. Curcumin also helped reduce mucus hypersecretion. More intriguingly, OVA-induced NF-κB p65 activation was abolished. Overall, the findings of this study suggest that curcumin significantly reduced OVA- and IL-4-induced airway inflammation and mucus hypersecretion via the PPARγ-dependent NF-κB signaling pathway in both lung and BEAS-2B cells. These conclusions appear to be consistent with previous findings indicating that NF-κB plays an important role in the regulation of inflammasome activation [67,68]. Taken together, these observations support curcumin’s role as an effective therapy for controlling the severity of chronic asthma through inflammasome regulation [69]. Different studies have demonstrated that tetrahydrocurcumin (THC), the major active metabolite of curcumin, or dexamethasone (DEX) could improve nasal symptoms and counteract pathological lungs alterations, which are consistent with the findings discussed above. THC and DEX have comparable therapeutic potential in various pathological contexts. Interestingly, when compared to monotherapy with either THC or DEX alone, the combination of both agents significantly reduced nasal rubbing frequency, mucus hypersecretion, and Th2 and Th17 cell count, and it reduced the related cytokines IL-4, IL-5, and IL-17involved in inflammasome activation and pyroptosis in pulmonary infectious diseases [70,71]. Further research is needed to improve curcumin’s low bioavailability and to investigate the use of dry powder inhalation for asthma to overcome solubility and dissolution issues. Taken together, most of the published data show that curcumin can alleviate inflammasome-associated diseases and their clinical manifestations when compared to commonly used drugs.

A large number of studies have shown that excessive inflammasome activation causes systemic and lethal inflammation in patients with viral infections. In particular, the NLRP3 inflammasome has been shown to play a critical role in the pathogenesis of viral diseases [72,73,74]. Recently, it was discovered that inflammasomes play an important role in the pathogenesis of emerging viral infections such as SARS-CoV-2. SARS-CoV-2 replication can be combined with both direct and indirect inflammasome activation. Coronavirus disease 2019 (COVID-19) is a devastating respiratory disease caused by SARS-CoV-2. Activation of the inflammasome in this context resulted in the generation of a severe cytokine storm, which causes serious complications such as acute respiratory syndrome (ARDS), acute pneumonia, and/or multiple organ dysfunction and ultimately death. Viral infection is followed by an inflammatory response that can cause tissue damage. The NLRP3 inflammasome has been reported to play an important role in COVID-19 pathogenesis [75]. A recent study investigating whether pre-existing asthmatic inflammation affected SARS-CoV-2-induced NLRP3 inflammasome activation in the mice’s lungs. Interestingly, it was found that after SARS-CoV-2 infection, the NLRP3 inflammasome components NLRP3, ASC, cleaved caspase-1, and IL-1β were significantly increased in Aspergillus fumigatus-challenged mice compared with control groups. These findings suggest that SARS-CoV-2 infection in severe asthma can be associated with COVID-19 worsening via NLRP3 inflammasome activation. As a result, the NLRP3 inflammasome may be considered as a potential target for early COVID-19 treatment in severe asthma [76]. Clinical data on the effectiveness of drugs used to treat COVID-19 is still limited and inconclusive. Curcumin has demonstrated an antiviral activity against a variety of viral diseases involving enveloped viruses, including SARS-CoV-2 [77].

Investigation of effective antiviral agents with low cytotoxicity remains of high priority. Curcumin has been shown to have antiviral activity against a variety of enveloped viruses, including SARS-CoV-2. This effect involves direct interaction with viral membrane proteins, disruption of the viral envelope, inhibition of viral proteases, and induction of a host response. Curcumin protects against lethal pneumonia and RDS via the NF-κB pathway, inflammasomes, IL-6 trans-signals, and the high mobility group box 1 (HMBG1) pathway (Figure 2). All the curcumin properties listed above make it an excellent and promising prophylactic therapeutic candidate for COVID-19 treatment in both clinical and public health settings [78]. Recent evidence from the literature has reported important crosstalk between the NRF2 and inflammasome signaling pathways. In this context, Nrf2-activating compounds inhibit inflammasomes and the resulting inflammation [79]. Given the pleiotropic effects of curcumin, it was hypothesized that curcumin could prevent SARS-CoV-2 entry, replication, and infection [80]. Furthermore, NRF2 agonists were shown to inhibit SARS-CoV-2 replication in lung cells, implying that NRF2 agonists are potential candidates. Given that curcumin is a promising NRF2 agonist capable of activating NRF2 pathways in the lungs of mice [81], curcumin can exert antiviral activity against SARS-CoV-2 by activating the NRF2 pathway. Curcumin has been shown to inhibit NLRP3 inflammasome activation, caspase-1 cleavage, and IL-1β secretion in PMA-induced macrophages. Curcumin’s signaling effects have been linked to TLR4/MyD88/NF-κB and P2X7R signaling pathways [82].

In line with what was described previously [25], following SARS-CoV-2 infection, NEK7-mediated activation of the NLRP3 inflammasome is coordinated by K^+^ efflux [73]. However, the molecular mechanisms are not yet fully understood. The inhibitory effects of curcumin against COVID-19 involving inflammasome activation are illustrated in Figure 2.

In agreement with the study by Saeedi-Boroujeni et al., recent findings reported that macrophages play a crucial role in the immunopathogenesis of COVID-19. Macrophages derived from inflammatory monocytes from patients with severe COVID-19 replace lung tissue-resident alveolar macrophages in the BAL fluid. Highly inflammatory macrophages were found in the lungs of these patients. This suggests that curcumin may be an effective anti-inflammatory compound in patients with severe COVID-19 [83]. In a recent study, curcumin was shown to have in vitro antiviral activity against SARS-CoV-2 at different concentrations using pre- and post-infection treatment strategies.

Curcumin also prevented SARS-CoV-2 delta variant infection and showed anti-inflammatory effects in PBMC challenged with SARS-CoV-2, modulating the expression of pro-inflammatory cytokines IL-1β, IL-6, IL-8, MCP-1, and INF-α. This suggests that regardless of the virus strain or variant, the viral replication cycle is inhibited [84]. Complementary research is required to establish its efficacy in human and animal models and to unravel its mechanisms of action. Curcumin’s anti-inflammatory properties in COVID-19 patients have been investigated in two separate studies. On the one hand, Valizadeh et al. found that nanocurcumin treatment increased mRNA levels and secretion of IL-1β, IL-6, TNF-α, and IL-18. The treatment with nanocurcumin significantly reduced IL-6 and IL-1β mRNA levels [85]. The second study found that Th17 cells, as well as serum levels of Th17-regulating factors such as IL-17, IL-21, IL-23, and GM-CSF were significantly reduced in patients with COVID-19 [86]. Other mechanisms involving the release of neutrophil extracellular traps (NETs), which occur during a regulated form of neutrophil cell death called NETosis, were described in patients with COVID-19. Indeed, following SARS-CoV-2 infection, NETosis and NET formation in both circulatory and infiltrating neutrophils resulted in extensive inflammation, pulmonary injury, and the formation of a typical COVID-19 thrombus [74,87]. Given the fact that NETosis and NET are widely recognized as mediators of pathophysiological abnormalities in SARS-CoV-2 infection, this axis remains critical in the pathogenesis of COVID-19. Hence, it is necessary to suppress the triggering factors involved in these processes, which could help reduce NETosis and prevent the negative consequences of NET formation.

Altogether, curcumin’s antiviral activity against enveloped viruses was attributed to a variety of mechanisms, including direct interaction with viral proteins, envelope disruption, inhibition of viral proteases, modulation of inflammatory responses, and modulation of host factors NF-κB, NRF2, and/or HMGB1 pathways. Furthermore, inhibiting the activation of NLRP3 or the formation of NETs by SARS-CoV-2 can be considered as a new potential strategy for the treatment of COVID-19 using natural compounds such as curcumin.

## 5. Curcumin, Inflammasome, and Arthritic Diseases

Arthritic diseases such as gout and rheumatoid arthritis (RA) are characterized by a loss of immunological tolerance to autoantigens, persistent overproduction of autoantibodies, and chronic inflammation [88]. Gout is caused by the accumulation of monosodium urate (MSU) crystals in joints, tendons, and other tissues, while RA is characterized by persistent synovitis and progressive cartilage, bone, and joint destruction [89]. The drugs used to treat these diseases have serious side effects and high costs [90]. As a result, safe alternative therapies, such as the use of natural compounds, could be useful [90]. Curcumin has been suggested for the treatment of arthritic diseases [63,91]. Chandran and Goel provided the first evidence for the safety of curcumin treatment in patients with active RA [92]. RA patients receiving curcumin (500 mg) had significantly reduced erythrocyte sedimentation rate (ESR), disease activity score 28 (DAS28), swollen joint count (SJC), tender joint count (TJC), visual analog scale (VAS) pain, and VAS activity, and C-reactive protein (CRP) significantly decreased only in the curcumin group [92]. In the same way, Amalraj et al. showed that curcumin (250 mg and 500 mg) significantly reduced ESR, CRP, VAS, SJC, TJC, rheumatoid factor (RF) values, DAS28, and American College of Rheumatology (ACR) responses in active RA patients compared to placebo; no side effects were reported [93].

Similarly, [94] it has been demonstrated that curcumin (250−1500 mg/day) reduced ESR and CRP in RA patients when compared with a control group. Furthermore, Pourhabibi-Zarandi et al. showed that curcumin (500 mg) significantly reduced homeostatic model assessment for insulin resistance (HOMA-IR), ESR, CRP, triglycerides, weight, body mass index, and waist circumference of women with RA compared with the placebo [95]. However, a low dose of curcumin nanomicelle (40 mg) administrated to RA patients three times a day for 12 weeks had no significant effect on ESR, DAS-28, TJC, and SJC when compared to the placebo group [95,96].

Curcumin (200 mg/kg) alleviated RA-induced inflammation and synovial hyperplasia in a rat collagen-induced arthritis (CIA) model by decreasing the levels of mTOR, p70S6K, and Akt1 and increasing the level of 4E-BP1, thus inhibiting the increased levels of IL-1β, TNF-α, MMP-1, and MMP-3 in the serum and synovium [97]. Consistently, [98] in the CIA rat model, it was found that curcumin (200 and 100 mg/kg) reduced the severity of arthritis and completely prevented joint histopathological changes (including edema, bone/cartilage destruction, synovial hyperplasia, and pannus formation). Similarly, Manca et al. confirmed the effect of curcumin in alleviating the severity of arthritis and synovial tissue damage by reducing infiltration of inflammatory cells and decreasing levels of IL-1, IL-6, and TNF-α in mice in a CIA model [99].

According to Xu et al., curcumin (50 mg/kg) was also effective in reversing the increase in arthritis scores, hind paw edema, and loss of appetite as well as suppressing the inflammatory response by reducing TNF-α, IL-6, and IL-17 and inhibiting the activation of the PI3K/AKT signaling pathway in a mouse CIA model [100]. Similarly, Xião et al. confirmed the effect of curcumin on alleviating the severity of arthritis and synovial tissue damage in a mouse CIA model by reducing inflammatory cell infiltration and decreasing levels of IL-1β, IL-6, and TNF-α [101]. Curcumin-loaded solid lipid nanoparticles (C-SLNs, 10 and 30 mg/kg) and 30 mg/kg (but not 10 mg/kg) of free curcumin significantly reversed joint hyperalgesia and decreased paw withdrawal threshold in a rat CIA model [102].

Furthermore, C-SLNs (10 and 30 mg/kg) and free curcumin (30 mg/kg) significantly reduced the increased number of leukocytes, serum MDA, nitrite, TNF-α, CRP, and CCP antibody levels and significantly increasing GSH, SOD activity, and catalase [102]. The NLRP3 inflammasome appears to be involved in the development of several arthritic diseases, including RA, ankylosing spondylitis, and gout [103,104,105]. This provides the first evidence that curcumin slows the progression of osteoarthritis disease by inhibiting the release of the NLRP3 inflammasome, resulting in the downregulation of inflammatory cytokines.

In a study conducted by Fan et al. in 2018 [106], hyaluronic acid/curcumin (HA/Cur) nanomicelle injected into RA rats reduced edema, friction between cartilage surfaces around joints, and expression of IL-1β, TNF-α, and vascular endothelial growth factor (VEGF). The NLRP3 inflammasome appears to play a role in the development of several arthritic diseases, including rheumatoid arthritis, ankylosing spondylitis, and gout [103,104].

Furthermore, Gu et al. [107] proposed that the curcumin analogue AI-44, by targeting cathepsin B and inhibiting NLRP3 inflammasome activation, is a novel drug candidate for the treatment of gouty arthritis. AI-44 inhibited the interaction of cathepsin B and NLRP3 to prevent NLRP3 inflammasome activation. Li et al. [108] have demonstrated that curcumin could alleviate MSU crystal-induced gouty arthritis by inhibiting NLRP3 inflammasome mediated by the NF-κB signaling pathway in both primary rat abdominal macrophages in 3-(4,5-dimethylthiazol-2-yl)-2,5-diphenyl tetrazolium bromide (MTT)- and MSU-induced gouty arthritis rat models. Curcumin also reduced MSU-induced inflammation by suppressing IκBα degradation, NF-κB signaling pathway activation, mitochondrial damage, and NLRP3 inflammasome activity [109].

It was reported that 75 mg/kg of the combination of tetramethylpyrazine, resveratrol, and curcumin (TRC) improved histopathological assessment by decreasing inflammatory cell infiltration, synovium thickness, and volumetric increase in the synovia space. All TCR formulations have been non-toxic [110]. Curcumin’s effect in combination with various anti-inflammatory drugs and other bioactive natural compounds was also studied as an alternative approach to treating RA. The TRC at 50 and 75 mg/kg (but not 25 mg/kg) decreased serum levels of TNF-α, IL-1β, and IL-6 in the CIA rat model [109]. The 75 mg/kg TCR combination improved histopathological assessment by decreasing inflammatory cell infiltration, decreasing synovium thickness, and reducing the volumetric increase in the synovia space. All TCR formulations were non-toxic [110,111], demonstrating that Cur (Cureit/Acumin^TM^) in combination with vitamin D_3_ and an omega-3 fatty acid (O3FA)-enriched diet significantly enhanced the suppression (>2-fold compared to Cur) of TNF-α, IFN-γ, and MCP-1 in a mouse CIA model.

Liposome-encapsulated dimethyl curcumin (Lipo-DiMC) has also been suggested as a promising treatment for RA [112]. Compared with untreated CIA rats, intra-articular injections of Lipo-DiMC (400 µL) relieved paw-swellings and almost reduced the increase in neutrophils, lymphocytes, monocytes, eosinophils, and leukocytes in blood. It also inhibited DPPI over-activity and MMP-2/9 overexpression in blood. Using LPS-stimulated HIG-82 synovial cells, He et al. [113] discovered that MSC-derived exosomes loaded with curcumin (Curc-Exos, 1:4 ratio) increased cytotoxicity and apoptosis while reducing the expression levels of anti-apoptotic proteins IAP1 and IAP2 and inflammatory mediators such as IL-6, TNF-α, MMP1, and PGE2 compared with the free curcumin treatment. Manca et al. discovered that curcumin-loaded hyalurosomes decreased the production of anti-apoptotic proteins IAP1 and IAP2, reduced the production of IL6, IL15, TNF-α, and ROS, and stimulated the production of IL-10 in RA-FLS cells [99]. Similarly, Xiao et al. [101] demonstrated that curcumin promoted apoptosis while inhibiting RA-FLS cell growth, migration, and invasion via the linc00052/miR-126-5p/PIAS2 axis.

The main cellular fraction within the synovium’s intimal layer is fibroblast-like synoviocytes (FLS). During the development of RA, inflammation-related cytokines such as IL-1β, IL-6, or TNF-α activate FLS, transforming their phenotype into cancer-like cells known as RA-FLS cells. RA-FLS cells release cytokines, chemokines, and adhesion molecules that exacerbate auto-inflammation and contribute to joint tissue destruction [114].

Curcumin inhibited TNF-α-induced proliferation, migration, and invasion of SV-40-transformed MH7A, and fibroblast-like synoviocyte (RA-FLS) cells while also promoting cell apoptosis [100,115]. In another RA in vitro model, curcumin (at 10 µM) reduced the survival of synovial sarcoma SW982 cells as well as MMP1 gene expression and TNF-α protein production [114]. Wang et al. [98] found that curcumin (0–10 μmol/L) also had a cytotoxic effect on mouse macrophage-like RAW 264.7 cells and could significantly inhibit the degradation of IκBα in LPS-induced RAW264.7 macrophages and in the synovium of the joints of CIA rats [98]. Taken together, in vitro, in vivo, and clinical studies confirmed curcumin’s promising effects for attenuating inflammation and oxidative stress, alleviating the patient’s pain, protecting joints from damage, and improving the quality of life of RA patients without side effects. Curcumin encapsulation in nanoparticles or in combination with MSCs showed an effective and promising strategy for increasing curcumin bioavailability and efficacy in RA. In addition, new curcumin formulations and administration routes (such as oral and topical) have been proposed as effective strategy for treating RA.

## 6. Conclusions

In this review, we have focused on the therapeutic effects of curcumin administration in various inflammatory diseases, focusing on its functional mechanisms in preventing activation of the NLRP3 inflammasome. Future research will address the clinical application of curcumin to treat diseases in which the inflammasome is activated.

## Figures and Tables

**Figure 1 molecules-28-00742-f001:**
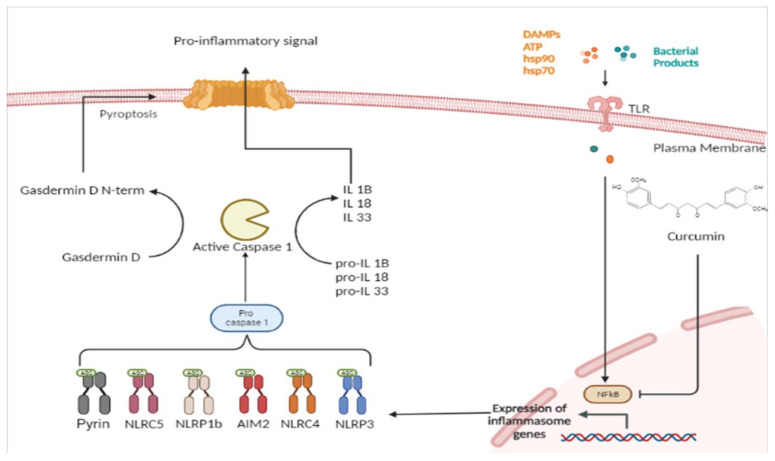
Curcumin inhibition of canonical inflammasome.

**Figure 2 molecules-28-00742-f002:**
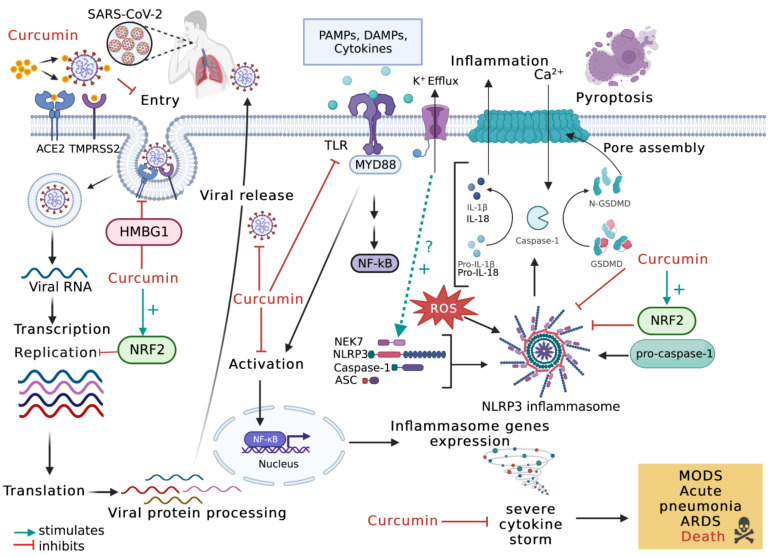
Potential inflammasome-related mechanisms by which curcumin is effective against COVID-19. Inflammasome activation is mediated by potassium ion efflux and by NEK7 and NLRP3 oligomerization The antiviral effect of curcumin against SARS-CoV-2 starts by preventing its entry. Curcumin inhibits NF-κB via the inhibition of different pathways. The binding of DAMPs, PAMPs, and cytokines to TLR leads to its activation and NF-κB translocation to the nucleus, which results in the expression of inflammasome genes. NF-κB activation induces the formation of a protein complex known as the inflammasome, which can lead to cell death through pyroptosis, a pathway to cell death mediated by the activation of caspase-1. However, curcumin can inhibit inflammasome formation by the inhibition of NF-κB. Curcumin acts as a potential inhibitor of NF-κB pathway activation. Curcumin induces antiviral responses by positively regulating NFR2 and negatively regulating HMGB1. Following SARS-CoV-2 infection, curcumin modulates responses by inhibiting cytokine responses and oxidative stress, which prevents the progression towards ARDS, acute pneumonia, MODS, and even death. TMPRSS2, transmembrane protease; PAMPs, pathogen-associated molecular pattern; DAMPs, damage-associated molecular patterns; NF-κB, nuclear factor kappa B; ACE2, angiotensin-converting enzyme 2; HMBG1: High mobility group box 1; NEK7: NIMA Related Kinase 7. Created with BioRender.com.

## Data Availability

Not applicable.

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
