# Peer review of "The Effects of Curcumin on Inflammasome: Latest Update"

_molecules, 2023, doi:10.3390/molecules28020742_

Round 1
Reviewer 1 Report
The topic of the research article is of great interest. However, I would not recommend publishing the article in its current format as it requires lots of improvement. The main drawbacks of this manuscript
Below are several specific comments.
1. The English writing should be further improved, as there are many grammatical or typing errors.
2. The whole manuscript is mixed between American English and British English. The authors even use British English or American English. For consistency, consider replacing it with the American English spelling.
3. Some references not according to journal instructions
4. Reference no. 86 should be 2021 not 2012
5-The authors used around 115 references as it's well-known that a review paper should use the last 5 years of literature, to obtain the most relevant information. The percentages of the last 5 years of references were 53.9 % (62 references) while old references were. 46.1 % (53 references).
6. Some types, grammar, and style errors need to be corrected listed below
line 18 changes protein to proteins
line 23 adds a comma before and
line 25 adds a comma after review
line 40 removes the comma after longa
line 48 adds the or an before excessive
line 55 adds a comma before and
line 58 changes for to of
line 72 changes resultings to resulting
line 93 adds a comma after hsp90
line 99 removes the comma after Caspase-1
line 100 adds a comma before and
line 102 changes are to is
line 109 changes activate to activates
line 111 changes (IALRs) to (ALRs)
line 114 adds and before AIM2
line 114 changes carboxyl terminal to carboxyl-terminal
line 115 changes recognizes to recognize
line 116 changes NATCHT to NACHT
line 126 adds are before known
line 126 changes activates to activate
line 135 adds a before cellular
line 144 changes tumour to tumor
line 156 changes defence to defense
line 156 changes Bacillus anthraces to Bacillus anthracis
line 158 changes toxin to toxins
line 161 changes best characterized to best-characterized
Line 170 changes are to is
Line 172 adds the before cytosolic
Line 173 adds are before somehow
Line 177 adds the ore an before optimal
line 180 changes signalling to signaling
Line 189 adds the before PYD
Line 189 changes pro-caspsase-1 to pro-caspase-1
line 209 changes import to important
line 211 changes in to on
lines 212 and 216 removes the comma before activated
line 222 changes channel, resulting in an increase in to channels, increasing
line 228 adds the before brain
line 256 removes of
line 264 adds a comma after specifically
line 266 changes nanotheranostic to nano theranostics
line 276 removes the comma after NLRP3
line 277 adds the before NLRP3
line 278 changes speck like to speck-like
line 288 adds been before fully
line 294 removes of
line 298 adds a comma before and
line 303 changes on to for
line 304 changes property to properties
line 333 removes the before acute
line 342 changes signalling to signaling
line 365 changes that that to that
line 374 changes modulate to modulates
line 375 changes regulate to regulates
line 390 changes signalling to signaling
line 393 removes the after support
line 400 changes agent’s to agents
line 414 adds an before important
Line 418 adds an before inflammatory
Line 418 changes tissues to tissue
Line 421 changes mice to mice’s
Line 423 adds a comma before and
Line 426 removes as
Line 427 removes a before severe
Line 436 adds are before listed
Line 439 remove an
line 442 adds a comma before and
lines 448 and 449 change signalling to signaling
line 462 changes to unravel to unraveling
line 464 changes nanocurcumin to nano curcumin
line 468 removes the comma before were
Line 476 adds a comma before and
Line 477 adds the before nucleus
Line 477 changes resulting to results
Line 478 changes in duces to induces
Line 478 adds space between TheNF-κB
Line 484 adds a comma before and
Line 489 adds a comma before and
Line 494 adds a comma before and
Line 495 changes cost to costs
Line 496 changes compounds to compound
Line 497 changes arthritics to arthritic
line 519 adds the before CIA and adds a comma after model
line 527 adds a comma before and
line 531 changes Curcumin loaded to Curcumin-loaded
Line 549 remove the comma after activation
line 561 changes were resulted to have resulted in
line 563 changes treat to treating
line 564 adds a comma before and
line 567 changes formulation to formulations
line 567 changes demonstrated to demonstrating
Line 576 who discovered it???
Line 577 changes ration to ratio
Line 580 changes curcumin loaded to curcumin-loaded
Line 587 removes the after into
line 589 remove the before joint
line 590 adds a comma before and
line 593 adds a comma before and
line 598 changes patients to patient’s
line 599 changes effect to effects
line 603 changes effective strategy to to an effective strategy for
line 606 changes prevent to preventing
line 607 changes address to clinic application on to address the clinical application of
line 607 changes for treat to to treat
Reviewer 2 Report
Major points
1. The inflammasome section is not properly described. Firstly, the inflammasome classification, according to the current literature classifies inflammasome in canonical, non-canonical and alternative inflammasome. The author should discuss all three types of inflammasome in pathophysiology contexts and their molecular mechanisms.
2. The authors report that, “based on the nucleotide-binding oligomerization domain, the inflammasome can be classified into four groups (NODs, NLRPs, NLRC4 and NLRC5)”, however, NOD domains include NLRPs as well as NLRCs and do not constitute a different group. Moreover, Pyrin inflammasome is not mentioned at all – “pyrin is quite distinct among inflammasome-forming receptors and only shares the PYD domain with other inflammasomes” [Function and mechanism of the pyrin inflammasome, Rosalie Heilig and Petr Broz, 2017, European Journal of Immunology, https://doi.org/10.1002/eji.201746947], therefore is classified as an additional independent inflammasome. In my opinion the author should discuss more deeply these aspects to give comprehensive description of the field of the inflammasome as well as their involvements in diseases.
3. At today the non-canonical inflammasome is better characterize than before, although not as good as the canonical inflammasome, however the authors mention the activation of caspase-11 as a marker for the non-canonical one. Caspase-11 is murine caspase not human; its human homologous are caspase-4 and -5 which are not mentioned at all.
4. The 2 steps that lead to the fully activation of the inflammasome, which are common to all the canonical inflammasomes are not described. For instance, the engagement of TLRs for the “priming step” is not described; LPS, which primes the most well characterized inflammasome (NLRP3) is not even mentioned in the inflammasome paragraph.
The molecular mechanism of activation of the speck formation is missed. All the inflammasomes form a single speck, however, NLRP3 speck is located at the perinuclear region and co-localize with MTOC, which makes this inflammasome different from others, where the single speck is within the cytosol w/o a specific structural cytosolic marker. Moreover, the fully activation of NLRP3 needs TGN, ox-mtDNA, NEK7 and cytoskeleton rupture which makes this inflammasome unique. All these aspects should be discussed more in detail.
The authors report that exists three models for NLRP3 inflammasome activation, however, all these activators bring to ROS production. The presence of ROS as intermediated is not a separated model but belong to all the mechanisms described today. Moreover, the bacterial toxin, nigericin, which represent one of the most potent stimuli, is missing. Here, several references are missing too.
The description of GSDMD should be put in the inflammasome section.
Finally, the role of the axis NLRP3-NETs-SARS-CoV2 should be also mentioned as a new potential target.
Minor points
In Fig.1 Active Caspase 1 is spelled wrongly; NLRC5 is not drawn
The single speck is 0.8-1 um not 1 uM
In Fig.2 the NLRP3 inflammasome has been drawn with NEK7 protein, which is needed for the fully activation, but it is not reported in the text; NLRP3, the 3 is missing in the inactive form
Round 2
Reviewer 1 Report
The authors have appropriately corrected the manuscript according to the comments
Reviewer 2 Report
I believe the manuscript has been sufficiently improved to warrant
publication in Molecules